# Study of Flebogrif^®^—A New Tool for Mechanical Sclerotherapy—Effectiveness Assessment Based on Animal Model

**DOI:** 10.3390/nano11020544

**Published:** 2021-02-21

**Authors:** Zbigniew Rybak, Maciej Janeczek, Maciej Dobrzynski, Marta Wujczyk, Albert Czerski, Piotr Kuropka, Agnieszka Noszczyk-Nowak, Maria Szymonowicz, Aleksandra Sender-Janeczek, Katarzyna Wiglusz, Rafal J. Wiglusz

**Affiliations:** 1Department for Experimental Surgery and Biomaterials Research, Wroclaw Medical University, Bujwida 44, 50-345 Wroclaw, Poland; zbigniew.rybak@umed.wroc.pl (Z.R.); maria.szymonowicz@umed.wroc.pl (M.S.); 2Department of Biostructure and Animal Physiology, Wroclaw University of Environmental and Life Sciences, Kozuchowska 1, 51-631 Wroclaw, Poland; maciej.janeczek@upwr.edu.pl (M.J.); albert.czerski@upwr.edu.pl (A.C.); piotr.kuropka@upwr.edu.pl (P.K.); 3Department of Pediatric Dentistry and Preclinical Dentistry, Wroclaw Medical University, Krakowska 26, 50-425 Wroclaw, Poland; maciej.dobrzynski@umed.wroc.pl; 4Institute of Low Temperature and Structure Research, PAS, Okolna 2, 50-422 Wroclaw, Poland; m.wujczyk@intibs.pl; 5Department of Internal Medicine and Clinic of Diseases of Horses, Dogs and Cats, Wroclaw University of Environmental and Life Sciences, pl. Grunwaldzki 47, 50-366 Wroclaw, Poland; agnieszka.noszczyk-nowak@upwr.edu.pl; 6Department of Periodontology, Wroclaw Medical University, Krakowska 26, 50-425 Wroclaw, Poland; aleksandra.sender-janeczek@umed.wroc.pl; 7Department of Analytical Chemistry, Wroclaw Medical University, Borowska 211 A, 50-566 Wroclaw, Poland; katarzyna.wiglusz@umed.wroc.pl

**Keywords:** mechanochemical ablation, Flebogrif^®^, endovenous procedure, animal model, experimental surgery

## Abstract

Sclerotherapy is the chemical occlusion of vessels using an intravenous injection of a liquid or foamed sclerosing agent that is used in the therapy of blood and lymphatic vessels malformations in the young, and for spider veins, smaller varicose veins, hemorrhoids and hydroceles in adults. This study aimed to assess the effectiveness of mechanosclerotherapy of venous veins with a new device—Flebogrif^®^—based on an animal model. The experiment was performed on nine Polish Merino sheep weighing 40–50 kilograms. The animals were anesthetized intravenously. The material was divided into three groups: two experimental (1 and 2) and control (3) group. The first experimental group was treated with the use of Flebogrif^®^ and a sclerosant simultaneously, while only Flebogrif^®^ was used in the second experimental group. Flebogrif^®^ was applied into the lateral saphenous vein of both pelvic limbs. The vessel wall thickness was estimated at four points of the histological image in mm (V1, V2, V3, V4). For one month, the animals were euthanized, and the occlusion rate of the treated veins and changes in the vein wall were determined. Histological slides were analyzed under a light microscope and histometry of the vein wall was performed. The Shapiro–Wilk test and the quantity of the investigated parameter groups allowed for using a non-parametric method at four points to compare thickness measurements (the Mann–Whitney test), with *p* < 0.05. The Mann–Whitney test indicated statistically significant differences between both experimental groups. The results obtained from morphometrical and histological analysis showed better results in the first experimental group than those of the second experimental group. Finally, statistical analysis revealed significant differences between the both the experimental group and control group in morphological analysis. The achieved results allowed us to conclude that the simultaneous use of Flebogrif^®^ and a sclerosant yielded better results of vein lumen reduction than the use of Flebogrif^®^ alone.

## 1. Introduction

The purpose of sclerotherapy is the chemical occlusion of insufficient vessels and varicose veins of superficial and perforating parts of the diseased venous network using an intravenous injection of a liquid or foamed sclerosing agent. Sclerosing solutions may be categorized by the endothelial injury mechanism they cause into detergent (e.g., sodium morrhuate, ethanolamine oleate, sodium tetradecyl sulfate, polidocanol), osmotic (e.g., sodium salicylate, sodium chloride, and sodium salicylate, dextrose, glycerin), or chemical (e.g., polyiodinated iodine) [1]. Commonly used sclerotherapy agents include lauromacrogol (polidocanol), sodium tetradecyl sulfate, and dextrose with hypertonic saline (25% dextrose, 10% sodium chloride) [2]. Lauromacrogol [C_12_H_25_O(CH_2_CH_2_O)_9_H] is a non-ionic, long-chain surfactant, polyethylene glycol ether of lauryl alcohol (see Figure 1).

Lauromacrogol-based sclerosants are being sold and trademarked in Europe [3]. It has been reported that lauromacrogol is successfully used in vessels ranging in diameter from 1 to 5 mm, depending on the active agent concentration in the applied sclerosing solution [1]. Additionally, lauromacrogol is used to treat skin conditions, including eczema or cowhage [4,5]. Lauromacrogol may be used in the sclerotherapy in the form of liquid or foam. However, the sclerosant damages the venous endothelium and possibly the tunica media of the vein wall. After successful sclerotherapy, the veins are transformed into fibrous cords in the long term treatment [6,7,8,9]. The purpose of sclerotherapy is not to achieve thrombosis of the vessel per se, which may recanalize, but to transform it into a fibrous tissue cord definitively. The optimal result of this procedure is similar to the surgical removal of a varicose vein [10], which involves consuming the entire refluxing vein by macrophages. In recent years many devices have been designed to improve this procedure, referred to as instruments for mechanosclerotherapy [11,12,13]. ClariVein is the most popular among those. This tool works by causing a mechanical injury to endothelium with a vibrating wire and simultaneous injection of a sclerosant [14,15,16]. Therefore, the aim of this study was the presentation of Flebogrif^®^, a new sclerotherapy device. Its concept is similar to ClariVein but more straightforward and cost-effective [11] (see Figure 2). 

As far as new techniques are concerned, there are still many unknowns related to the long-term results of treatment. The average practitioner in phlebology world-wide is interested in achieving the best results. The general concept of the hybrid procedure—related to abolishing retrograde flow in the truncal superficial veins of the lower extremity combining mechanical abrasion of the endothelium and medial part of the vein wall with sclerotherapy—theoretically could give better results than sclerotherapy alone.

Flebogrif^®^ (Balton, Poland) is a novel mechanochemical ablation device for treatment of saphenous vein insufficiency, either great saphenous or small saphenous. It combines venous wall damage performed by radial retractable cutting hooks together with endothelial chemical ablation through sclerosant injection of polidocanol foam. The objective of this study was to evaluate Flebogrif’s efficacy in terms of size of damage of vein wall. 

Endovascular techniques for the treatment of saphenous vein insufficiency have been increasing in number over the last years. Among them, endovenous laser ablation (EVLA) and radiofrequency ablation (RFA) are now considered first-choice treatments, according to the latest guidelines for truncal ablation [10]. Those procedures require expensive generators and special fibers. Looking for more simple, convenient and inexpensive technics mechanochemical ablation (MOCA) as a hybrid procedure seems to be reasonable [17].

Nowadays, two devices are accessible on the market that are commercially patented as Clarivein^®^ (Merit Medical, South Jordan, UT, USA) [18] and Flebogrif^®^ (Balton, Poland), respectively. The endothelial and medial part of the venous wall mechanical damage is supported to enhance the penetration of the sclerosant in the vessel wall and the subsequent vasoconstriction as proven in ex vivo and animal models. Thus, it is reasonable to assume that occlusion rates would be higher than with mechanical or chemical ablation treatments alone [19]. 

The immediate impact of sclerosant on the vein wall was assessed in a number publications [20,21].

## 2. Materials and Methods

### 2.1. The Analysis of Physicochemical Properties of Lauromacrogol

The study of physicochemical properties of lauromacrogol was performed using optical spectroscopy techniques, including UV-Vis-NIR absorption, infrared and micro-Raman spectroscopy. The absorption spectrum was determined using a Cary 5000 spectrophotometer (Santa Clara, CA, USA) with the 175–3300 nm range on the quartz plate. The spectrum obtained was in the 300–2500 nm range with a resolution of 0.5 nm. A Nicolet iS10 FT-IR spectrometer (Waltham, MA, USA) equipped with an automated beam splitter exchange system (iS50 ABX containing a DLaTGSKBr detector) with a HeNe laser as an IR radiation source and built-in all-reflective diamond ATR module (iS50 ATR), Thermo Scientific Polaris™, was used for the infrared spectrum measurements. The spectra detected were in the range of 400–4000 cm^−1^ with a frequency resolution of 4 cm^−1^. A potassium bromide (KBr) plate was used for the measurement. The Micro-Raman spectrum was determined with a Renishaw InVia micro-Raman system (Wotton-under-Edge, Gloucestershire, UK) equipped with a Leica DM 2500 M microscope and a CCD camera for detection (3500–200 cm^−1^). The spectrum recorded was in the range 50–3300 cm^−1^ when excited at 514 nm.

### 2.2. The Course of the Experiment

The material was divided into three groups: (1) animals were treated with the use of Flebogrif^®^ and a sclerosant in form of foam; (2) only Flebogrif^®^ was used; (3) the sclerosant in form of foam was used in the case of the control group. The study was approved by a local Ethics Committee for Animal Experimentation (No. 129/2010). The experiment was performed on nine Polish Merino sheep, weighing 40–50 kilograms. The premedication features included detomidine hydrochloride (Domosedan^®^, Orion Corporation, Espoo, Finland) in IM injection 40 ug kg/BW. The animals were anesthetized intravenously using propofol (Scanofol^®^, Norbrook Laboratories Ltd, Newry, Northern Ireland). The catheter used in this experiment was developed in collaboration with the medical equipment manufacturer Balton Sp. z o.o. Warsaw, Poland. The design of the catheter is based on a typical 5F 100 cm long single-channel diagnostic catheter. A metal shank, to which are attached five thin, curved, springy wires with sharpened ends, is introduced into the catheter. After being pushed out of the catheter, these wires deploy into a cat’s claw pattern. The purpose of these ‘claws’ is to carve deep bruises on the internal wall of the vein when the catheter is pulled out of the vessel. The thickness and elasticity of the wires making up the claws were adjusted to make it possible to deeply bruise the endothelium and the median part while minimizing the risk of vein wall perforation. The distance between the fully expanded claws of Flebogrif^®^ is about 20 µm [11]. Flebogrif^®^ was applied into the lateral saphenous vein of both pelvic limbs. For sclerotherapy, lauromacrogol at a concentration of 3% was used (Kreussler-Pharma, Wiesbaden, Germany). In each case, 5 mL of a foam form of sclerosant was injected simultaneously following retraction of Flebogrif^®^. The foam was prepared according to Tessari method. It was mixture of one part (1 mL) of lauromacrogol and four parts (4 mL) of the room air.

### 2.3. Histological Evaluation

One month after surgery the animals were euthanized and the samples of the veins were taken for histological examination. The material was fixed in 4% buffered formaldehyde pH 7,2, dehydrated in alcohol series and embedded in paraffin, and then cut with a Leica RM 2045 (Leica Camera AG, Wetzlar, Germany) rotation microtome into 7 µm sections. The samples were stained with hematoxylin-eosin (H-E) and Masson–Goldner methods. The material was observed under a light microscope (Axio Scope A1, Carl Zeiss, Jena, Germany). The vessel wall thickness was estimated at four points of the histological image in mm (V1 - 4). The measurements were taken using the Axio-vision 4.8 software (Carl Zeiss, Jena, Germany). The statistical analysis of results was performed using the IBM SPSS 23.0 software. The Shapiro–Wilk test and the quantity of the investigated parameter groups allowed for a non-parametric method to be used at four points of thickness measurement comparison (Mann–Whitney test), with *p* < 0.05.

## 3. Results

### 3.1. The Analysis of Physicochemical Properties of Lauromacrogol

Lauromacrogol transmits the UV-Vis region of the spectrum (see Figure 3) [22,23]. Relevant absorption lines are prominent in the NIR region of the spectrum [23].

A Fourier-transform infrared (FT-IR) spectrum has been recorded for lauromacrogol; it is presented in Figure 4. The observed bands can be associated with the molecule’s characteristic functional groups. The spectrum can be divided into two distinct spectral regions, the first 3650–2650 cm^−1^, and the second 1550–500 cm^−1^. The high-frequency region consists of two bands, a broadband at 3500 cm^−1^, which is attributed to the stretching mode of the O–H group, and a sharp, split band at 3050–2650 cm^−1^,which can be assigned to the sp^3^ C–H stretching mode. The second region consists of bands associated with CH_2_ (1466 cm^−1^), CH_3_ (1348 cm^−1^), and C–O–H (1252 cm^−1^) bending modes (δ)and with the highest intensity C–O (1116 cm^-1^) stretching mode (ν). Further, for lower frequencies observed are C–C (947 cm^−1^) stretching (ν), C–H (843 cm^−1^) out of plane bending (δ_oop_), and CH_2_ (723 cm^−1^) long-chain band– rocking mode (γ). Obtained data are in correlation with these already published by Smarandache et al. [22]. 

Lauromacrogol was investigated via micro-Raman spectroscopy for additional information on the vibrational modes of the compound (Figure 5). The mode with the highest intensity, at around 2700–3000 cm^−1^, can be assigned to methyl C–H symmetric and antisymmetric stretching [24]. The second most intense vibration mode at 1438 cm^−1^ can be attributed to H–C–H methyl out-of-plane deformation. Skeletal vibrations can be observed in the form of –(CH_2_)_n_ in-phase twist mode at 1300–1240 cm^−1^ and C–C skeletal stretching mode at 1130–1050 cm^−1^ and 900–800 cm^−1^. Additionally, visible modes at around 900–800 cm^-1^ can also be attributed to symmetric stretching of C–O–C [25].

### 3.2. Histological Results

The pilot study demonstrated the high efficacy of mechanochemical ablation to treat incompetent saphenous veins. The small number of minor complications typical for standard chemical ablation confirms the high level of safety of this technique. Because both the device and method are innovative, the next research step likely to be taken will involve a direct evaluation of Flebogrif^®^ compared to other already established surgical techniques. Considering the good results of ClariVein, our mechanochemical ablation concept seems to be correct [13,14,15,16]. The mechanism of action of Flebogrif^®^ has a solid pathophysiological basis. There was an excellent response of the vein wall in the first group of animals (treated with Flebogrif^®^ and a sclerosant) (see Figure 6 and Figure 7). 

Histological evaluation showed increased fibroblast activity in the tunica intima and media and almost no response in adventitia in 1st group. Endothelium was restored in the entire surface of the examined vessels. The newly synthesized collagen fibers between the smooth muscle cells caused the loss of the tunica media integrity, therefore in the newly formed tissue single smooth muscles may be observed (Figure 7). In this area small capillary vessels may be observed. This is due to vascularization of this area induced by injury. No signs of inflammation in the vein wall were observed in all analyzed groups. Less intense changes were observed in 2nd group. Here, the tunica intima and media were less active and no significant changes were observed in wall structure. The adventitia remains intact in both experimental groups.

Using Flebogrif^®^ alone, the vessel wall reaction was less pronounced (see Figure 8 and Figure 9). A comparison of the results of both experimental groups and the control group indicated statistical significance. The experiment based on an animal model has confirmed the clinical outcomes reported by the authors using Flebogrif^®^ together with a sclerosant.

### 3.3. The Morphometrical Analysis 

The Mann–Whitney test results of the vein wall thickness indicated statistically significant differences between both experimental groups and the control group are presented in Table 1.

The results showed statistically significant differences between the thickness of the vein wall 1 and the control group (see Table 2). The results of the first experimental group were higher than those of the control group. 

Finally, statistical analysis indicated significant differences between the second experimental group and the control group (see Table 3). The average values for all groups are shown in Figure 10.

Morphometrical analysis indicate significant differences in vein wall thickness between the experimental groups and control group. The thickest wall was observed in 1st group. This means, that tissue response was observed after simultaneous use of Flebogriff and the sclerosing agent. 

This suggests, that use only of the sclerosing solution may induce changes in the vein wall, however, much more intensive reactions may be obtained after use of sclerosing solutions with Flebogriff.

## 4. Discussion

Sclerotherapy of insufficient superficial veins of the legs is the most frequent procedure in phlebology worldwide. Lauromacrogol is the number one among sclerosant agent used for the above purposes. Foam sclerotherapy offers many advantages but it is not free of complications, comprising thrombophlebitis, inflammation of the skin and hyperpigmentation. All these complications seem to be related to thrombosis of the treated vein. Furthermore, the high postprocedural efficacy of foam sclerotherapy, which is reported as an occlusion of the treated vein, results from immediate thrombosis of such a vein, which precedes a slow process of inflammatory driven fibrosis evoked by chemical irritation [11]. Thrombosis is a reversible process—hence the high rate of late recanalization of previously occluded veins, even as high as 20–30% after 12 months [26]. The mechanical injury of the vein wall appears to be the initial and pivotal event that initiates damage of the endothelium and enables penetration of the sclerosant into deeper layers of the venous wall. Mechanochemical obliteration should be more effective in comparison with standard chemical ablation, since the former should result in more severe and diffuse inflammatory process that leads to fibrosis of the vein. Ciostek et al. performed a pilot study of the Flebogrif^®^ catheter in clinical settings in the years 2011–2013. He included 40 patients presenting with class C2–C6 of chronic venous disease and with incompetence of the great or small saphenous veins demonstrated by duplex Doppler. Thirty-nine patients completed the study. The efficacy of the procedure—defined as occlusion of the treated saphenous vein—assessed at follow-up visits was: after one month 97.4% (38/39), after 3 months 94.9% (37/39), after 6 months 89.7% (35/39) and after 12 months 89.7% (35/39) [11].

Lauromacrogol is an active agent in applied sclerosing solutions and foams. It has been investigated for its physicochemical properties. Optical spectroscopy techniques including absorption, FT-IR, and Raman have been used to examine in detail the interaction of light with the applied agent’s chemical bonds. This compound belongs to the class of detergent sclerosing solutions that are nonionic compounds. It consists of a polar hydrophobic part (dodecyl alcohol) and a polar hydrophilic part (polyethylene-oxide chain) that is esterified. In solution, lauromacrogel is associated as macromolecules through electrostatic hydrogen bonding between the H^+^ atom of the OH^−^ group in one molecule, and the free electron-pair of an oxygen atom of a second molecule. This bonding results in the formation of a network. Thus, the sclerotherapeutic activity results from this double hydrophobic and hydrophilic action. The optimal efficacy of the compound coincides with the highest concentration that still permits the existence of non-aggregated molecules. The randomized control clinical study performed by Rasmussen revealed that endovenous laser ablation, radiofrequency ablation, foam sclerotherapy, and surgical stripping of great saphenous varicose veins in humans are of no significance [15]. The above-mentioned vein corresponds to the medial saphenous vein in animals. Both medial and lateral saphenous veins in domestic sheep are similar in size [27].

For the past years, open surgery has been the recommended and predominantly used procedure for treating varicose veins. About 20 years ago, the development of a minimally invasive correction of primary superficial venous reflux in patients with a chronic venous disease of the lower limbs by endovenous techniques provided a patient-friendly means to treat this disorder in an office-based procedure with the occlusion of the saphenous veins and varicosities, including radiofrequency ablation, endovenous laser ablation, and sclerotherapy. In addition, a type of surgery preserving the great saphenous vein (GSV) was developed. A new mechanochemical device (ClariVein) has been designed to minimize the adverse effects of both endothermal ablation and ultrasound-guided sclerotherapy to treat saphenous incompetence, incorporating the benefits of each. The advantages of this hybrid system are said to include standard percutaneous access, endovenous treatment, local anesthesia but without the need for tumescent anesthesia, and a short procedure time. Since the system does not use thermal energy, the potential for nerve injury is minimal. The mechanochemical method achieves venous occlusion utilizing a wire rotating within the vein lumen at 3500 rpm, injuring the intima to allow for better efficacy of the sclerosant. These two modalities, mechanical and chemical, give the results of venous occlusion equal to endothermal methods. Similar results were obtained by Ammollo et al. [19] and Iłżecki et al. [28]. The histological findings are similar to those obtained by other authors. As a result of the damage to endothelial cells, the increased activation of fibroblasts results in occlusion of the vein wall, based upon two mechanisms. 

The first mechanism is related to the contraction of smooth muscles present in the tunica media; in the second mechanism, activated fibroblasts synthesize collagen fibers which causes the thickening of the tunica intima. This leads to the occlusion of the vessels. One month after surgery, increased vascularization of this area is visible in the tunica intima. 

## 5. Conclusions

By the means of the absorption, infrared and Raman spectroscopy techniques, the lauromacrogol (polidocanol) has been precisely characterized. The detailed physicochemical analysis was the basis for carrying out further experiments. The above outcomes allowed us to conclude that the simultaneous use of Flebogrif^®^ and a sclerosant (lauromacrogol) yielded better results of vein lumen reduction than the use of Flebogrif^®^ alone. The preliminary study showed no direct damage done to the vein wall by Flebogrif and a slight increase in wall diameter. In combination Flebogrif^®^ + sclerosant was observed to increase the connective tissue of the intima.

## Figures and Tables

**Figure 1 nanomaterials-11-00544-f001:**
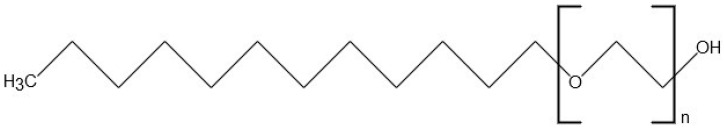
Skeletal formula of lauromacrogol (polidocanol) single molecule.

**Figure 2 nanomaterials-11-00544-f002:**
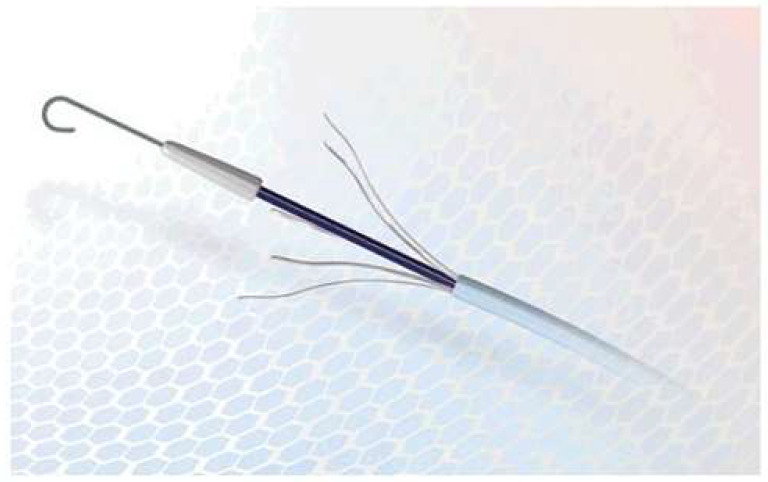
Flebogrif^®^—a new tool for mechanosclerotherapy.

**Figure 3 nanomaterials-11-00544-f003:**
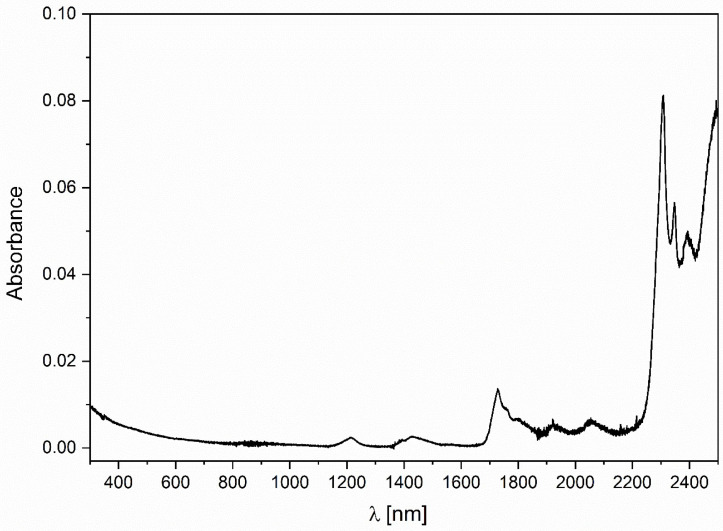
The absorption spectrum of lauromacrogol.

**Figure 4 nanomaterials-11-00544-f004:**
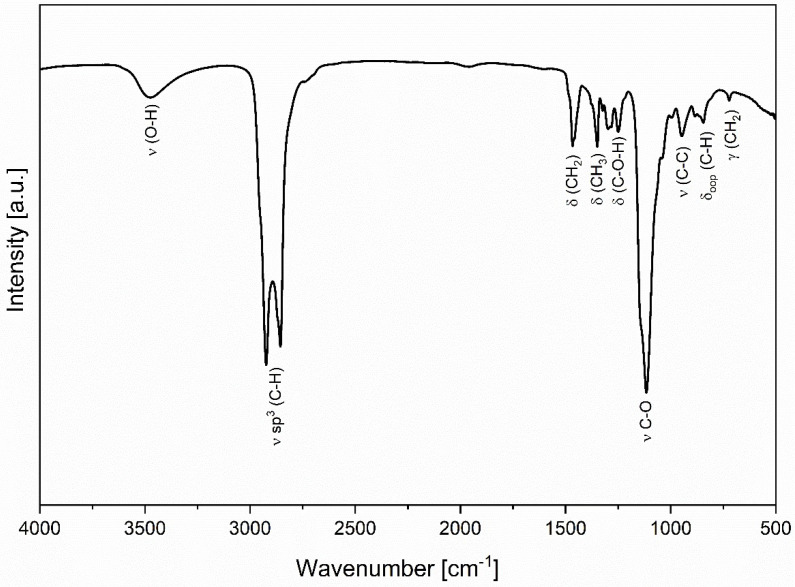
Fourier-transform infrared (FT-IR) spectrum of lauromacrogol.

**Figure 5 nanomaterials-11-00544-f005:**
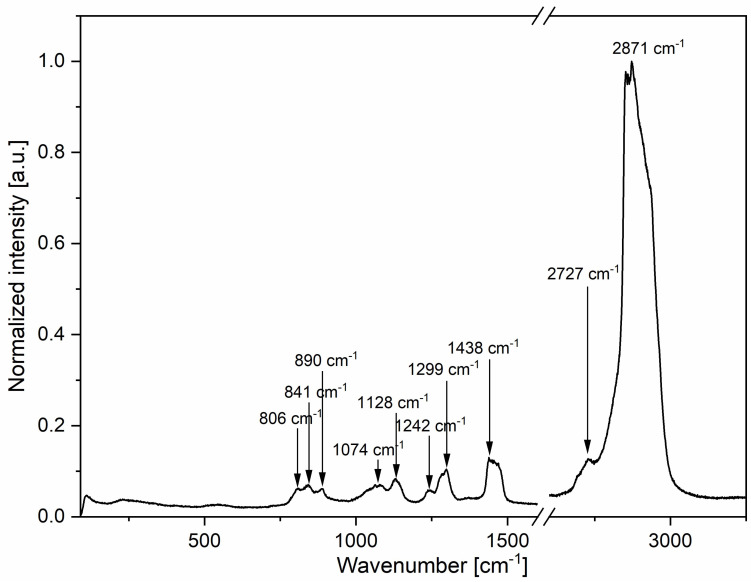
Micro-Raman spectrum of lauromacrogol excited at 514 nm.

**Figure 6 nanomaterials-11-00544-f006:**
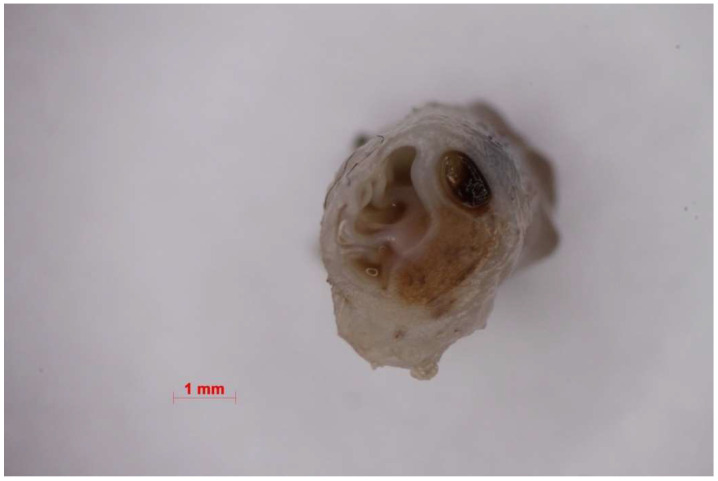
Macroscopic picture of the vein lumen treated with Flebogrif^®^ + sclerosant.

**Figure 7 nanomaterials-11-00544-f007:**
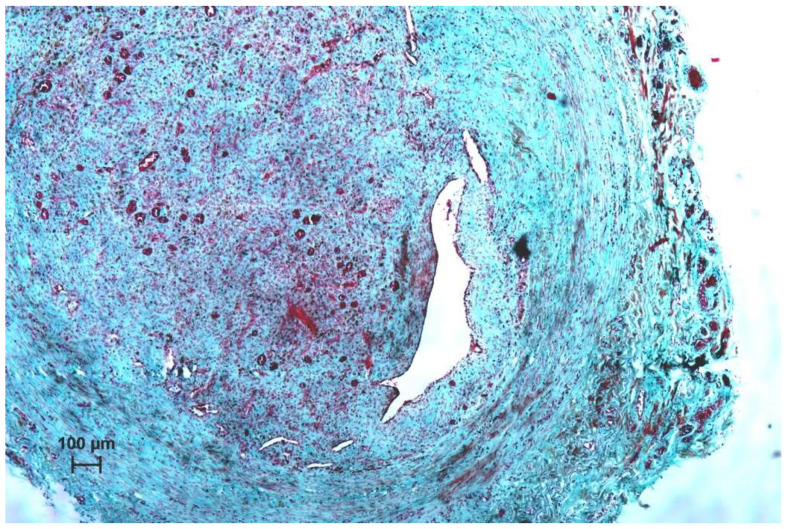
Histological picture of the vein treated with Flebogrif^®^ + sclerosant, Masson–Goldner staining, ×10 mag.

**Figure 8 nanomaterials-11-00544-f008:**
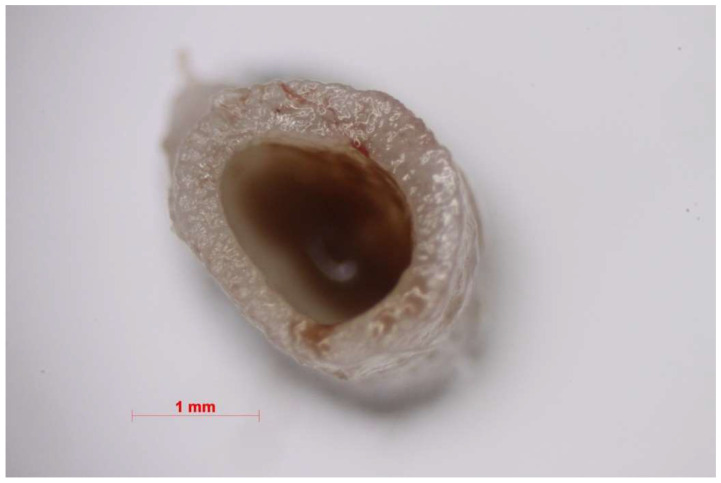
Macroscopic picture of the vein lumen treated with Flebogrif^®^.

**Figure 9 nanomaterials-11-00544-f009:**
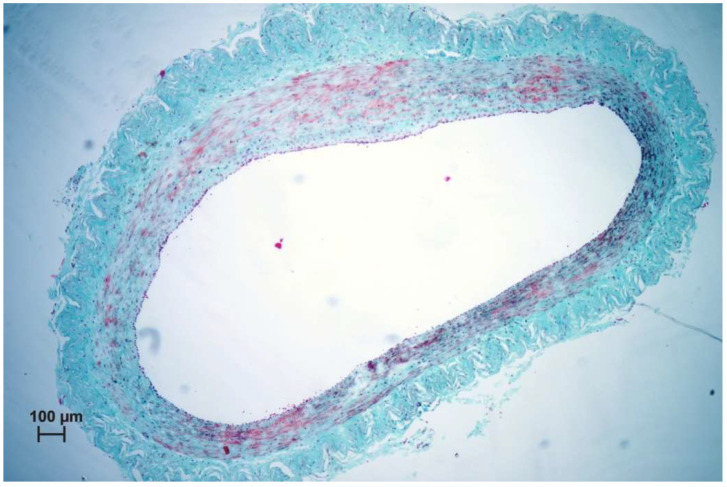
Histological picture of the vein treated with Flebogrif^®^– Masson–Goldner staining, ×5 mag.

**Figure 10 nanomaterials-11-00544-f010:**
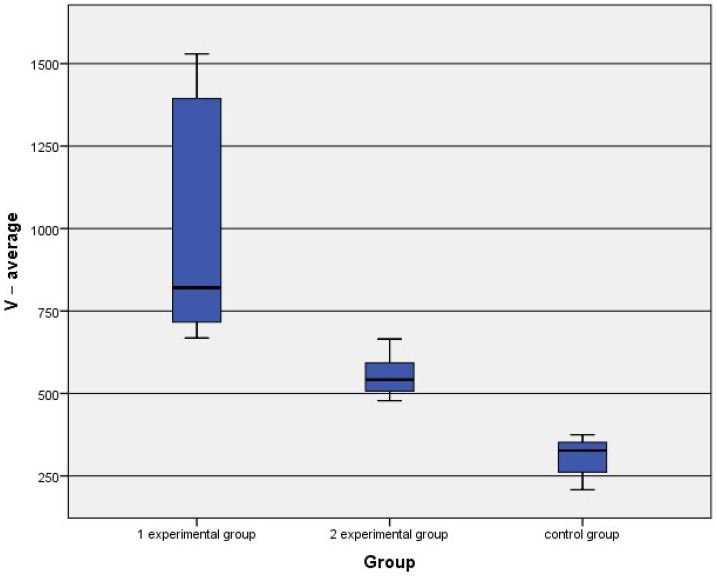
Comparison of the average values of wall thickness between both experimental groups and the control group.

**Table 1 nanomaterials-11-00544-t001:** Mann–Whitney test: group 1–2.

	V-1	V-2	V-3	V-4	V-Average
**Mann–Whitney U**	25.000	32.000	13.000	4.000	0.000
**Wilcoxon W**	61.000	68.000	49.000	40.000	36.000
**Z**	−1.333	−0.711	−2.399	−3.199	−3.554
***p***	0.183	0.477	0.016	0.001	0.000

**Table 2 nanomaterials-11-00544-t002:** Mann–Whitney test: group 1–3.

	V-1	V-2	V-3	V-4	V-Average
**Mann–Whitney U**	8.000	4.000	0.000	2.000	0.000
**Wilcoxon W**	53.000	49.000	45.000	47.000	45.000
**Z**	−3.021	−3.348	−3.674	−3.511	−3.674
***p***	0.003	0.001	0.000	0.000	0.000

**Table 3 nanomaterials-11-00544-t003:** Mann–Whitney test: group 2–3.

	V-1	V-2	V-3	V-4	V-Average
**Mann–Whitney U**	11.000	7.000	0.000	18.000	0.000
**Wilcoxon W**	56.000	52.000	45.000	63.000	45.000
**Z**	−2.406	−2.791	−3.464	−1.732	−3.464
***p***	0.016	0.005	0.001	0.083	0.001

## Data Availability

Not applicable.

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
