# Peer review of "Study of Flebogrif®—A New Tool for Mechanical Sclerotherapy—Effectiveness Assessment Based on Animal Model"

_nanomaterials, 2021, doi:10.3390/nano11020544_

Round 1

Reviewer 1 Report

The authors present and evaluated a new mechanical device to treat abnormal veins. A lot is focussed on the presentation of the evaluation of  a sclerosend agent. It is not clear to me why this agent is so extensively evaluated and described as the focus should be on the new device. I would suggest that this part is taken out or shortened considerable.

Also I am wondering why a group of only treatment with the foam was not included. I would suggest the authors to extend the experiments with such a group. It might well be that the foam alone has a similar effect as the device. 

In the title I would remove the word brand new and replace it with just new.

Author Response

Dear Editor,

We would like to express our sincerest gratitude to the Reviewers for their enormous efforts in criticizing the manuscript. We have taken into account all raised question here follows the detailed answers to Reviewers. Moreover, all changes we have made to the original manuscript and marked in the red colour in the text.

REVIEWER 1.

  1. The authors present and evaluated a new mechanical device to treat abnormal veins. A lot is focussed on the presentation of the evaluation of  a sclerosend agent. It is not clear to me why this agent is so extensively evaluated and described as the focus should be on the new device. I would suggest that this part is taken out or shortened considerable.

Answer: We agree with the Reviewer and this part has been rewritten.

  1. Also I am wondering why a group of only treatment with the foam was not included. I would suggest the authors to extend the experiments with such a group. It might well be that the foam alone has a similar effect as the device. 

Answer: It is well known in the literature that  polidocanol as a sclerosant agent in the treatment of venous disorders is very effective. However,  if it is compared with the other sclerosant agents in long-term observations there is observed relatively high percentage of the recanalization in such treatments. In such cases, flebologists try to combine the procedures like Flebogrif and sclerosant, so-called a hybrid procedure.

  1. In the title I would remove the word brand new and replace it with just new.

Answer: The “Brand” has been removed from the title of the manuscript.

Reviewer 2 Report

This is a manuscript that presents an idea that is potentially interesting but in which the message is completely lost due to the very poor understanding of scientific publication. 

ABSTRACT: An abstract should include 'background', 'objective/hypothesis', 'methods', 'results' and finally a 'conclusion'. The provided synopsis has no structure whatsoever. It also includes unnecessary information, such as line 31 - 'The animals were anesthetized intravenously', which is information that should not be part of the summary. There are other examples as well... 

INTRODUCTION: The first line of the introduction sets up the writing as it does not include the pertinent information regarding the pros or cons of using sclerotherapy. The rest is a compilation of (again) unnecessary information (for example, line 63 where the authors describe the trademarks or the sclerosants sold in Europe). The first figure is also an example of unnecessary information as it does not any important information. 

Normally an introduction is written to introduce the information that follows. In this example, very little of what follows is actually mentioned in this introduction. 

MATERIALS & METHODS: This section needs to be subdivided by using headings to avoid ending up with this type of poorly structured text. The description also has to follow the experimental events. A general guide is to write this section in a way that allows a reader, unfamiliar with the techniques, to be able to replicate the acquired data.  

RESULTS: The result section is very poorly written and starts off with information that should have been included in the introduction. It is impossible to get an appreciation for the results based on the writing. There is no correlation with the techniques and what is presented seems to be taken out of context. There is also lots of unnecessary information. For example, why is Figure 4 in the result section? Why are the results only presented as tables? What does the statistical difference represent? 

DISCUSSION: The discussion is filled with repetition and text that should have been placed in the introduction (for example line 180-200). This is not a properly written introduction but rather a repetition of the results as it does not put the authors data in context. There needs to be a reflection and links to already published studies. The section also includes figures (Figure 3-6) that do not add any important information. 

Overall, this is a very poorly written article. This reviewer strongly suggest that the authors find an older, more experienced researcher as a mentor. This individual should be able to provide guidance related to everything from how to design a research study to scientific writing.  

Author Response

Dear Editor,

We would like to express our sincerest gratitude to the Reviewers for their enormous efforts in criticizing the manuscript. We have taken into account all raised question here follows the detailed answers to Reviewers. Moreover, all changes we have made to the original manuscript and marked in the red colour in the text.

REVIEWER 2.

This is a manuscript that presents an idea that is potentially interesting but in which the message is completely lost due to the very poor understanding of scientific publication. 

  1. ABSTRACT: An abstract should include 'background', 'objective/hypothesis', 'methods', 'results' and finally a 'conclusion'. The provided synopsis has no structure whatsoever. It also includes unnecessary information, such as line 31 - 'The animals were anesthetized intravenously', which is information that should not be part of the summary. There are other examples as well... 

Answer: The Abstract has been rewritten as well as the unnecessary information have been removed.

  1. INTRODUCTION: The first line of the introduction sets up the writing as it does not include the pertinent information regarding the pros or cons of using sclerotherapy. The rest is a compilation of (again) unnecessary information (for example, line 63 where the authors describe the trademarks or the sclerosants sold in Europe). The first figure is also an example of unnecessary information as it does not any important information. Normally an introduction is written to introduce the information that follows. In this example, very little of what follows is actually mentioned in this introduction. 

Answer: The Introduction has been rewritten according to the Reviewer’s suggestion.

  1. MATERIALS & METHODS: This section needs to be subdivided by using headings to avoid ending up with this type of poorly structured text. The description also has to follow the experimental events. A general guide is to write this section in a way that allows a reader, unfamiliar with the techniques, to be able to replicate the acquired data.  

Answer: The Materials & Methods has been rewritten according to Reviewer’s suggestion.

  1. RESULTS: The result section is very poorly written and starts off with information that should have been included in the introduction. It is impossible to get an appreciation for the results based on the writing. There is no correlation with the techniques and what is presented seems to be taken out of context. There is also lots of unnecessary information. For example, why is Figure 4 in the result section? Why are the results only presented as tables? What does the statistical difference represent? 

Answer: The Results has been rewritten according to Reviewer’s suggestion.

  1. DISCUSSION: The discussion is filled with repetition and text that should have been placed in the introduction (for example line 180-200). This is not a properly written introduction but rather a repetition of the results as it does not put the authors data in context. There needs to be a reflection and links to already published studies. The section also includes figures (Figure 3-6) that do not add any important information. 

Answer: The Discussion has been rewritten according to Reviewer’s suggestion.

  1. Overall, this is a very poorly written article. This reviewer strongly suggest that the authors find an older, more experienced researcher as a mentor. This individual should be able to provide guidance related to everything from how to design a research study to scientific writing.

Answer: It is difficult to find an older and more experienced researcher in this field of medicine. However, it has been found a compromise among the authors and we have rewritten this part of the manuscript according to Reviewer’s suggestion.

Round 2

Reviewer 2 Report

Better version

Still missing the part in the text where the authors explain why this is important

Author Response

Dear Editor,

We would like to express once again our sincerest gratitude to the Reviewers for their enormous efforts in criticizing the manuscript. We have taken into account all raised question here follows the detailed answers to Reviewer. Moreover, all changes we have made to the original manuscript and marked in the red colour in the text.

REVIEWER 2.

Better version

Still missing the part in the text where the authors explain why this is important

Answer: We would like to thank the Reviewer once again for drawing our attention to this aspect. There is a part showing and explaining why this hybrid procedure is important. The Introduction has been rewritten according to the Reviewer’s suggestion.
